# NOAH: BENCHMARKING NARRATIVE PRIOR DRIVEN HALLUCINATION AND OMISSION IN VIDEO LARGE LANGUAGE MODELS

## ABSTRACT

Video large language models (Video LLMs) have recently achieved strong performance on tasks such as captioning, summarization, and question answering. Many models and training methods explicitly encourage continuity across events to enhance narrative coherence. While this improves fluency, it also introduces an inductive bias that prioritizes storyline consistency over strict grounding in visual evidence. We identify this bias, which we call *narrative prior*, as a key driver of two errors: hallucinations, where non-existent events are introduced or existing ones are misinterpreted, and omissions, where factual events are suppressed because they are misaligned with surrounding context. To systematically evaluate narrative prior–induced errors, we introduce NOAH, a large-scale benchmark that constructs composite videos by inserting clips from other sources into target videos. By varying semantic similarity and insertion position, our benchmark enables controlled and scalable analysis of narrative priors. We design one captioning task with tailored metrics and three QA tasks—Existence, Temporal, and Narrative—yielding more than 60K evaluation samples. Extensive experiments yield three key findings: (i) most Video LLMs exhibit hallucinations and omissions driven by narrative priors, (ii) the patterns of these errors vary across architectures and depend on event similarity and insertion position, and (iii) reliance on narrative priors intensifies under sampling with fewer frames, amplifying errors when event continuity is weak. We establish NOAH as the first standardized evaluation of narrative prior–induced hallucination and omission in Video LLMs, providing a foundation for developing more reliable and trustworthy models. Our benchmark and code are available at https://anonymous550520.github.io/.

## 1 INTRODUCTION

Video large language models (Video LLMs) (Maaz et al., 2024; Lin et al., 2024) have recently achieved strong performance on video understanding tasks such as captioning (Krishna et al., 2017), summarization (Ma et al., 2002), and question answering (Yang et al., 2003). However, they face two critical challenges: *hallucinations*—generating scenes, objects, or events absent from the video or misinterpreting those that are present—and *omissions*—suppressing factual events despite their presence. Both issues undermine the reliability of Video LLMs in real-world applications.

To address these challenges, several benchmarks have been proposed. Motion-oriented benchmark evaluates errors in perceiving visual movements (Kong et al., 2025). Event-level benchmarks assess semantic mistakes in "who did what and where" (Zhang et al., 2024b; Wang et al., 2024; Bae et al., 2025). Temporal benchmarks probe reasoning about event order, duration, and timing (Choong et al., 2024; Li et al., 2025). Beyond hallucination, ARGUS (Rawal et al., 2025) evaluates omission, measuring whether models overlook factual events depicted in the video.

Despite these efforts, hallucinations and omissions induced by *narrative priors* remain underexplored. We define a narrative prior as an inductive bias of Video LLMs to favor coherent storylines over strict grounding in visual evidence by integrating context across consecutive events. Recent advances in Video LLMs increasingly strengthen such priors by leveraging global context during training to capture inter-event dependencies and enhance narrative coherence Zhang et al. (2024e;a). While such priors improve fluency, they frequently lead to two failure modes: (i) hallucinations, where models introduce or reinterpret events to preserve coherence, and (ii) omissions, where mod-

Figure 1: Examples of hallucinations and omissions induced by narrative priors, generated by BLIP-3-Video (Ryoo et al., 2024). (a) A hallucinated caption is generated to maintain coherence between distinct events. (b) An event distinct from the others is omitted to preserve narrative continuity.

els suppress factual events that are misaligned with the surrounding context. Figure 1 illustrates these cases: (a) a resort commercial where the model hallucinates a non-existent transition event, and (b) a vacuum cleaner commercial where a factual noise-meter event is omitted because it disrupts the established narrative flow.

To systematically evaluate and analyze hallucinations and omissions induced by narrative priors, we introduce NOAH (Narrative prior-induced Omission and Hallucination benchmark). We carefully design NOAH to (i) isolate errors that arise specifically from narrative priors and (ii) provide a scalable framework for controlled evaluation. To this end, we construct composite videos from an event-annotated dataset, *i.e.*, ActivityNet-Captions (Krishna et al., 2017), by inserting an event-level clip from another source into a target video. By varying semantic similarity (high, medium, low) and insertion position (start, middle, end), we generate nine controlled variants per video, yielding 9,000 composite videos from 1,000 original videos.

Using these videos, NOAH evaluates models on one captioning task with tailored metrics and three QA tasks, all designed to measure model performance under narrative priors. Through the captioning task, we evaluate the extent to which models hallucinate or omit events in a video, under narrative priors. Through the QA tasks, we evaluate how much a model can (i) identify the presence of inserted clips (*Existence QA*), (ii) reason about the correct temporal order of inserted clips relative to surrounding events (*Temporal QA*), and (iii) distinguish plausible but false narrative events from factual events (*Narrative QA*). Altogether, NOAH provides over 60K evaluation samples.

Our comprehensive analysis on NOAH yields three insights: (i) most Video LLMs exhibit substantial hallucinations and omissions driven by narrative priors, (ii) the patterns of these errors vary across model architectures and depend strongly on both the semantic similarity between the inserted clip and the target video and the insertion position, and (iii) reliance on narrative priors intensifies when fewer frames are sampled, amplifying errors under weaker event-to-event continuity.

In summary, our contributions are threefold:

- **New perspective on hallucination and omission in Video LLMs**: We identify narrative priors as a previously underexplored cause of hallucinations and omissions, providing a novel lens for understanding the limitations of current Video LLMs.
- **Benchmark for narrative prior–induced errors**: We introduce NOAH, a large-scale benchmark that systematically constructs controlled composite videos to evaluate narrative prior-induced hallucinations and omissions.
- **Comprehensive evaluation and analysis**: We design one captioning task with tailored metrics and three QA tasks, and provide the first in-depth study of narrative prior–induced hallucinations and omissions, analyzing both their severity and their variability across model architectures, insertion settings, and frame sampling.

## 2 RELATED WORK

**Video Large Language Models.** Video LLMs extend the capabilities of large language models to video understanding, enabling tasks such as captioning, summarization, and question answering (Zhang et al., 2023; Maaz et al., 2024; Ma et al., 2024). Early approaches adapt image-based models by pooling frame-level features or projecting temporal embeddings, while more recent systems such as InternVideo2.5 (Wang et al., 2025), LongVILA (Chen et al., 2025), and MA-LMM (He

Figure 2: Overview of data construction. Candidate clips are ranked by CLIP cosine similarity, with high-, medium-, and low-similarity clips inserted at the start, middle, or end of each target video, yielding $3 \times 3 = 9$ composite variants per video to study narrative prior–induced errors.

et al., 2024) expand context windows to hundreds of frames to support long-span reasoning. Beyond architectural scaling, several methods explicitly promote narrative continuity. Shot2Story (Han et al., 2025) integrates shot-level annotations for coherent storytelling, MM-Narrator (Zhang et al., 2024a) employs multimodal memory for long-form narration, and ReVisionLLM (Hannan et al., 2025) shows that temporal grounding benefits from maintaining continuity across distant events.

While these approaches improve global understanding by reinforcing coherence across events, they also introduce errors. Models often generate hallucinations—fabricating absent events or misinterpreting actual ones—and omissions, where factual events are suppressed if they are misaligned with the surrounding context. In contrast to prior work that emphasizes coherence, our benchmark is the first to systematically evaluate hallucination and omission induced by *narrative priors*.

**Hallucination Benchmarks for Video LLMs.** One of the biggest concerns about the reliability of Video LLMs is hallucination—the generation of content inconsistent with visual evidence. Using a question-answering (QA) setup, VideoHallucer (Wang et al., 2024) and EventHallusion (Zhang et al., 2024b) probe object- and action-level hallucinations with adversarial questions, while UN-SCENE (Bae et al., 2025) diagnoses reliance on spurious action–scene correlations. These QA-based benchmarks provide valuable insights into local consistency but evaluate primarily at the clip level, offering a limited perspective on a multi-event video scenario. Vidhal (Choong et al., 2024), Vidhalluc (Li et al., 2025), and Vript (Yang et al., 2024) directly evaluate temporal misinterpretations by testing whether models correctly infer event order and duration. ARGUS (Rawal et al., 2025) represents a notable step forward, introducing the first systematic quantification of both hallucinations and omissions in open-ended captions.

Despite these advances, hallucinations and omissions induced by *narrative priors* remain underexplored. In this work, we introduce the first benchmark explicitly designed to evaluate narrative-driven hallucinations and omissions in a multi-event video setting.

## 3 NOAH BENCHMARK

We introduce NOAH, a benchmark for evaluating hallucinations and omissions induced by the *narrative priors* of Video LLMs. To isolate these errors from other causes, we construct composite videos by inserting a clip from another source into a target video. We control two key factors: (i) *semantic similarity* between the inserted clip and the target video, and (ii) *insertion position* within the target video. By systematically varying only these two dimensions, we could minimize potential confounding effects and can primarily attribute observed errors to narrative priors. Using these composite videos, NOAH evaluates models on one captioning and three QA tasks with tailored metrics, enabling a comprehensive assessment of Video LLMs reliability.

### 3.1 DATASET CONSTRUCTION

The overall data construction pipeline is illustrated in Figure 2. Based on two key intuitions, we create nine video variants to analyze hallucination and omission patterns induced by narrative priors.

**Intuition 1 (Semantic similarity).** *The degree of similarity between the target video and the inserted clip influences how strongly Video LLMs attempt to maintain narrative coherence.*

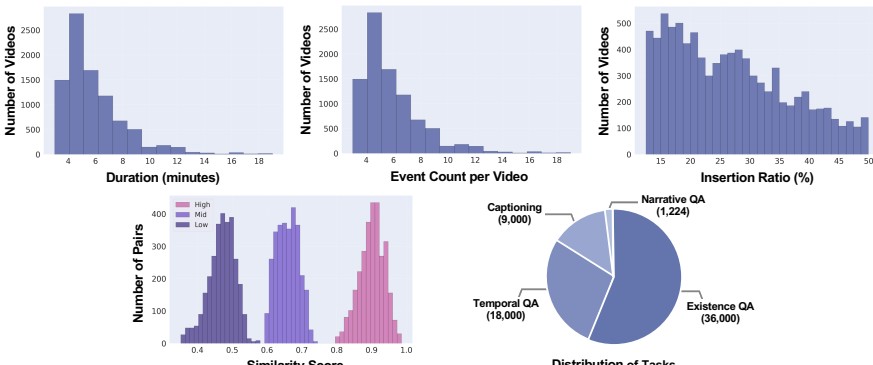

Figure 3: Dataset statistics. Top: distributions of duration (left), event count per video (middle), and insertion ratio—the proportion of an inserted clip within the composite video (right). Bottom: similarity distribution across high/mid/low groups (left) and the number of evaluation samples.

**Intuition 2 (Insertion position).** *The insertion position affects the patterns of hallucinations and omissions induced by narrative priors.*

We vary two factors at three levels each: (i) clip similarity to the target video (high, medium, low) and (ii) insertion position (start, middle, end), producing 9 variants per video. By fixing the original video and varying only the inserted event and its position, we establish a controlled setting to analyze patterns in the influence of narrative priors on model behavior. This clip insertion process across multiple target videos yields a large dataset for systematic analysis of hallucinations and omissions.

**Source Videos.** To construct NOAH, we curate videos from ActivityNet-Captions (Krishna et al., 2017), which provides multiple event-level annotations per video. We retain only videos with at least four non-overlapping events to ensure quality and diversity. For each target video, clips from other videos serve as insertion candidates, with durations restricted to [12.5%, 50%] of the target length to avoid trivial cases and to ensure that the inserted clip is not skipped during the frame sampling process of Video LLMs. Finally, we refine the original annotations using Gemini-2.0-Flash (Anil et al., 2023) to add fine-grained, visually grounded details, enabling reliable evaluation.

**Similarity Control.** To analyze how the effect of narrative priors varies with contextual plausibility, we select insertion clips based on their semantic similarity to the target video. We compute cosine similarity between temporally averaged CLIP (Radford et al., 2021) embeddings of the target and candidate clips, then rank candidates and select those with the highest, median, and lowest scores.

**Event Insertion Position.** We insert selected clips at the start, middle, or end of a target video—before the first frame, at the midpoint, or after the final frame—to examine how temporal context shapes model behavior induced by narrative priors. An insertion at the start might encourage the inserted clip as a premise for all subsequent content, an insertion in the middle requires coherence with both preceding and following events, and an insertion at the end primarily ties the narrative to prior context. Comparing performance across positions reveals whether hallucinations and omissions are driven more by preceding context, following context, or their interaction.

**Statistics.** Figure 3 summarizes key dataset statistics, including video durations, event counts, similarity distributions, and sample counts. We select 1K target videos from ActivityNet-Captions and construct 9K composite videos evenly distributed across three insertion positions and three similarity levels. The composite videos average 199 seconds in length (range: 14–728s) and contain 6.17 annotated events on average (range: 3–19). NOAH goes beyond single-event QA benchmarks, enabling controlled analysis of hallucination and omission in multi-event video settings.

## 3.2 TASK DESIGN

To evaluate *narrative prior induced* hallucinations and omissions of Video LLMs, we propose one captioning task and three question answering tasks as shown in Figure 4.

**Video Captioning under Narrative Priors.** Video captioning aims to generate factually faithful and comprehensive descriptions of video content, thus serving as a key measure of Video LLM reliability. To test this under conditions that induce *narrative prior*-driven errors, we design a captioning task using composite videos, where an event clip from another source is inserted into a target

Figure 4: Overview of four evaluation tasks. (1) Captioning assesses hallucination and omission in video descriptions; (2) Existence QA tests whether models correctly distinguish real inserted events from distractor events; (3) Temporal QA evaluates understanding of event order; (4) Narrative QA checks whether models reject fabricated but plausible events.

video. With this setup, we can evaluate whether models hallucinate visually ungrounded events or omit factual ones in pursuit of narrative coherence. To measure these effects, we introduce caption- and event-level hallucination and omission metrics in Section 4.1.

**Video Question Answering under Narrative Priors.** Beyond captioning, we design three binary QA tasks to evaluate whether models can (i) detect inserted events, (ii) infer temporal order of events, and (iii) determine whether a narratively plausible candidate event actually occurs, all under controlled conditions.

(**Existence QA**) This task evaluates whether a model can correctly verify the presence or absence of an inserted event, thereby testing its ability to ground answers in visual evidence rather than *narrative priors*. When the inserted clip is semantically similar to the target video, its presence may appear plausible; when dissimilar, context may suggest its absence. To ensure robustness against response bias, we generate paired questions in both affirmative and negative forms. For both forms, we construct questions using not only the inserted event but also a distractor event randomly sampled from ActivityNet-Captions dataset. In total, this design yields 36K samples, with four QA pairs generated per composite video.

(**Temporal QA**) This task tests whether a model can correctly identify the chronological order of the inserted event relative to its neighbors, assessing its ability to rely on the actual visual sequence rather than *narrative priors*. To ensure robustness against response bias, we also generate a pair of binary questions for each video, e.g., *"Before {reference event}, is the previous event {inserted event}?"* has the ground truth "Yes," while the corresponding *"After {reference event}, is the next event {inserted event}?"* has the ground truth "No." For insertions at the start or end (single-neighbor case), we create complementary question pairs against the adjacent event; for middle insertions (two-neighbor case), reference events are drawn equally from both preceding and succeeding neighbors. This design probes temporal reasoning in both forward and backward contexts, producing 18K samples in total—two QA pairs for each of the 9K videos.

(**Narrative QA**) This task evaluates whether a model can reject a *fabricated but narratively plausible* event triggered by an inserted clip. Unlike Existence QA, which checks visually grounded facts, Narrative QA tests whether models can distinguish actual events from coherent but false descriptions. To construct this task, we manually curate 306 fabricated events and pair them with factual events sampled from the same video. As in Existence QA, questions contrast factual events (ground truth: "Yes") with fabricated ones (ground truth: "No"); labels are inverted for the negative variant In total, this design yields 1,224 Narrative QA pairs.

Together, these tasks form a unified framework for evaluating how *narrative priors* induce hallucinations and omissions in both open-ended captioning and targeted QA tasks. Additional details on question templates and prompts are provided in the Appendix A.

## 4 EXPERIMENTS

### 4.1 EXPERIMENTAL SETUPS

**Evaluation Metrics.** For fine-grained evaluation, we design metrics tailored to captioning. For captioning, we introduce caption- and event-level measures of factual faithfulness, using an LLM-

based evaluator inspired by GAVIE (Liu et al., 2024a). The evaluator parses generated captions into events and compares them with ground-truth annotations to detect hallucinations and omissions. Further details, including prompts for event extraction, comparison procedures, and API inference costs, are provided in Appendix B.

(**Caption-level metrics**) We define the Caption Hallucination Rate (CHR) and Caption Omission Rate (COR) as the proportion of captions at least one hallucinated or omitted event:

$$\text{CHR} = |\mathbb{C}^{\text{hallu.}}|/|\mathbb{C}|, \quad \text{COR} = |\mathbb{C}^{\text{omiss.}}|/|\mathbb{C}|, \tag{1}$$

where $\mathbb{C}^{\text{hallu.}}$ and $\mathbb{C}^{\text{omiss.}}$ denote the set of captions with at least one hallucination or omission, and $\mathbb{C}$ is the set of all evaluated captions, respectively. These metrics offer an intuitive, user-facing measure: if a caption contains even a single hallucinated or omitted event, the entire description is deemed unreliable.

(**Event-level metrics**) While caption-level metrics are intuitive, they can obscure error patterns (Rohrbach et al., 2018): a single hallucination renders an entire caption unreliable, without indicating how many errors occurred or which events were affected. To provide finer granularity, we define event-level metrics: Event Hallucination Rate (EHR), Event Omission Rate (EOR), and Inserted Event Omission Rate (IEOR):

$$\text{EHR} = \frac{1}{|\mathbb{C}|} \sum_{c \in \mathbb{C}} \frac{|\mathbb{E}_c^{\text{hallu.}}|}{|\mathbb{E}_c^{\text{total}}|}, \ \text{EOR} = \frac{1}{|\mathbb{C}|} \sum_{c \in \mathbb{C}} \frac{|\mathbb{E}_c^{\text{omiss\_origin.}}|}{|\mathbb{E}_c^{\text{origin.}}|}, \ \text{IEOR} = \frac{1}{|\mathbb{C}|} \sum_{c \in \mathbb{C}} \frac{|\mathbb{E}_c^{\text{omiss\_insert.}}|}{|\mathbb{E}_c^{\text{insert.}}|}, \tag{2}$$

where $\mathbb{E}_c^{\text{total}}$ and $\mathbb{E}_c^{\text{hallu.}}$ are all and hallucinated events in a single caption $c$; $\mathbb{E}_c^{\text{origin.}}$ and $\mathbb{E}_c^{\text{omiss\_origin.}}$ are the original target video events and those omitted; and $\mathbb{E}_c^{\text{insert.}}$ and $\mathbb{E}_c^{\text{omiss\_insert.}}$ are the inserted events and those omitted. These metrics quantify the proportion of hallucinated (EHR) or omitted events—whether original (EOR) or inserted (IEOR)—and capture the cumulative severity of narrative prior–induced errors within captions, offering finer resolution than caption-level metrics.

(**QA metrics**) For Existence, Temporal, and Narrative QA tasks, we adopt a consistency-based evaluation protocol (Bae et al., 2025). Each Existence and Narrative sample consists of a pair of factual ("Yes") and counterfactual ("No") questions, controlling for polarity bias and enforcing balanced classes. A model is deemed correct only if it answers both consistently. Accuracy is reported as the proportion of correctly answered pairs over all instances.

**Video LLMs.** We evaluate a broad range of Video LLMs on NOAH, including both open- and closed-source models. Among open-source models, we cover diverse architectures and parameter scales (4B–72B), such as BLIP-3-Video (Ryoo et al., 2024), Video-LLaVA (Lin et al., 2024), LLaVA-NeXT-Video (Zhang et al., 2024d), LongVA (Zhang et al., 2024c), mPLUG-Owl3 (Ye et al.), LLaVA-OneVision (Li et al.), ST-LLM (Liu et al., 2024b), Video-LLaMA3 (Zhang et al., 2025), and Qwen2.5-VL (Bai et al., 2025). For closed-source models, we include Gemini 2.5 Flash (Comanici et al., 2025) and GPT-4o (2024-08-06) (OpenAI, 2024), representing state-of-the-art commercial models for large-scale video understanding.

**Implementation details.** To ensure reproducibility, we use official implementations with default settings (e.g., sampling strategies and the number of sampled frames) unless otherwise noted. For closed-source models, whose internal pipelines are not publicly disclosed, we follow their documented default configurations.

## 4.2 RESULTS

**Across all captioning metrics, Video LLMs exhibit a severe amount of errors.** Open-source models nearly always omit at least one original event per sample (COR $\approx$ 1) and hallucinate in most cases (CHR $\geq$ 0.7), with more than half of original (EOR $\geq$ 0.5) and inserted events omitted (high IEOR). These results indicate that narrative priors push models to preserve coherence at the expense of factual completeness. Closed-source systems perform relatively better—Gemini 2.5 Flash (CHR = 0.442, COR = 0.568) and GPT-4o (CHR = 0.494, COR = 0.864)—yet still hallucinate and omit nearly half of the events.

**Video LLMs fail on the binary QA tasks.** Failure patterns vary across models. Existence QA yields the highest scores overall, while LongVA performs comparatively better on Temporal QA than on other tasks. Surprisingly, GPT-4o records lower Temporal QA performance than several

| Models | # LLM Params | #F | Captioning | | | QA | | |
|---|---|---|---|---|---|---|---|---|
| | | | CHR / EHR (↓) | COR / EOR (↓) | IEOR (↓) | Exist. (↑) | Temp. (↑) | Narr. (↑) |
| **Open-source Video LLMs** | | | | | | | | |
| BLIP-3-Video | 4B | 8 | 0.888 / 0.493 | 0.994 / 0.598 | 0.692 | 0.364 | 0.101 | 0.142 |
| Video-LLaVA | 7B | 8 | 0.766 / 0.524 | 0.999 / 0.782 | 0.859 | 0.290 | 0.039 | 0.276 |
| LLaVA-NeXT-Video | 7B | 8 | 0.820 / 0.514 | 0.999 / 0.711 | 0.825 | 0.387 | 0.100 | 0.281 |
| LLaVA-OneVision | 7B | 8 | 0.709 / 0.278 | 0.979 / 0.483 | 0.438 | 0.685 | 0.486 | 0.387 |
| LongVA | 7B | 16 | 0.717 / 0.363 | 0.979 / 0.490 | 0.673 | 0.041 | 0.260 | 0.029 |
| mPLUG-Owl3 | 7B | 16 | 0.796 / 0.331 | 0.988 / 0.526 | 0.517 | 0.781 | 0.499 | 0.212 |
| ST-LLM | 7B | 64 | 0.835 / 0.428 | 0.996 / 0.584 | 0.686 | 0.250 | 0.060 | 0.283 |
| Video-LLaMA3 | 2B | 128 | 0.860 / 0.358 | 0.979 / 0.448 | 0.537 | 0.348 | 0.341 | 0.304 |
| Video-LLaMA3 | 7B | 128 | 0.751 / 0.274 | 0.966 / 0.420 | 0.468 | 0.460 | 0.460 | 0.154 |
| Qwen2.5-VL | 3B | 128 | 0.804 / 0.329 | 0.969 / 0.464 | 0.469 | 0.680 | 0.345 | 0.356 |
| Qwen2.5-VL | 7B | 128 | 0.670 / 0.223 | 0.928 / 0.373 | 0.351 | 0.970 | 0.600 | 0.536 |
| Qwen2.5-VL | 32B | 128 | 0.736 / 0.223 | 0.832 / 0.202 | 0.330 | 0.973 | 0.655 | 0.592 |
| Qwen2.5-VL | 72B | 128 | 0.705 / 0.247 | 0.935 / 0.393 | 0.344 | 0.937 | 0.682 | 0.796 |
| **Closed-source Video LLMs** | | | | | | | | |
| Gemini 2.5 Flash | - | 128 | 0.442 / 0.097 | 0.568 / 0.153 | 0.191 | 0.686 | 0.597 | 0.668 |
| GPT–4o | - | 128 | 0.494 / 0.136 | 0.864 / 0.263 | 0.401 | 0.703 | 0.148 | 0.471 |

Table 1: Evaluation results on NOAH. We report the performance of open- and closed-source Video LLMs on captioning and QA tasks.

open-source models, suggesting its imperfect temporal reasoning capability under narrative priors. These results highlight that narrative priors drive hallucinations and omissions in Video LLMs.

**Does an event insertion induce hallucinations and omissions driven by narrative priors?** To analyze narrative prior–driven errors, we perform a controlled evaluation. For each composite video $V_c$ in NOAH, we also consider its original video $V_o$ and the inserted clip $V_i$. Both $V_o$ and $V_i$ contain 8 sampled frames, while $V_c$ includes the same 8 frames from $V_o$ plus 1 additional frame from $V_i$.

| Models | EHR (↓) | | | EOR (↓) | | | IEOR (↓) | | | Exist. QA (↑) | | |
|---|---|---|---|---|---|---|---|---|---|---|---|---|
| | $V_o$ | $V_c$ | $\Delta(\%)$ | $V_o$ | $V_c$ | $\Delta(\%)$ | $V_i$ | $V_c$ | $\Delta(\%)$ | $V_i$ | $V_c$ | $\Delta(\%)$ |
| Video-LLaVA | 0.483 | 0.523 | + 8.28 | 0.764 | 0.792 | + 3.66 | 0.808 | 0.907 | +12.25 | 0.678 | 0.583 | –14.01 |
| LLaVA-NeXT-Video | 0.463 | 0.479 | + 3.46 | 0.699 | 0.678 | – 3.00 | 0.684 | 0.830 | +21.35 | 0.750 | 0.651 | –13.20 |
| LLaVA-OneVision | 0.249 | 0.266 | + 6.39 | 0.522 | 0.469 | –10.15 | 0.560 | 0.584 | + 4.29 | 0.937 | 0.830 | –11.42 |

Table 2: Controlled evaluation results for Video-LLaVA, LLaVA-NeXT-Video, and LLaVA-OneVision. EHR and EOR for captioning are computed by comparing $V_o$ and $V_c$, while IEOR for captioning and Existence QA are computed by comparing $V_i$ and $V_c$. $\Delta$ denotes the performance difference between $V_c$ and $V_o$ (or $V_i$).

**An event insertion induces hallucination and omissions of original events.** Table 2 compares $V_o$ and $V_c$ using EHR and EOR to measure hallucinations and omissions induced by event insertion. Across models, EHR consistently increases with insertion ($V_o \rightarrow V_c$), confirming that models become more prone to fabricating events. This provides direct evidence that *narrative priors actively drive hallucinations and omissions* beyond what visual input alone can explain. Interestingly, EOR sometimes improves when an event is inserted. For example, LLaVA-NeXT-Video and LLaVA-OneVision better understand original events in $V_c$ than in $V_o$, suggesting that depending on architecture, narrative priors may also reinforce understanding of original content.

**Video LLMs omit inserted events due to narrative priors.** In Table 2, we compare IEOR and Existence QA accuracy when a model uses $V_i$ and $V_c$ as an input. Models that correctly recognize an inserted clip in isolation ($V_i$) often omit it in the composite video ($V_c$), resulting in higher IEOR and lower Existence QA accuracy. In other words, Video LLMs tend to disregard an inserted clip to maintain narrative coherence.

Taken together, these findings show that hallucinations and omissions are driven by narrative priors of Video LLMs. Overall, the results provide strong evidence that narrative priors play a central role in shaping model behavior across both captioning and QA tasks.

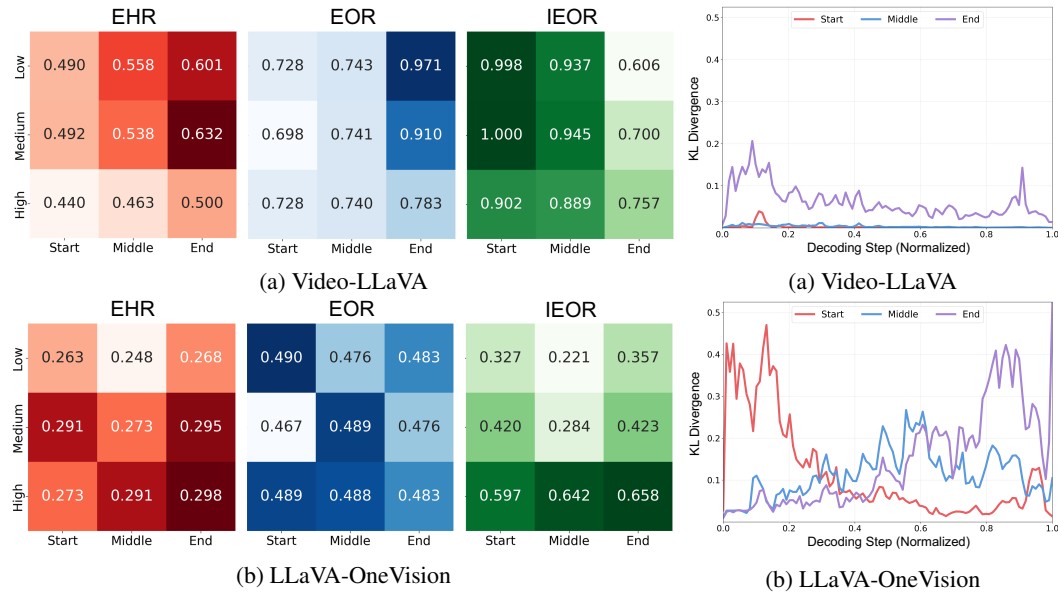

Figure 5: Heatmaps of captioning metrics across similarity and insertion position.

Figure 6: KL Divergence across insertion positions.

### 4.3 ANALYSIS

In this section, we conduct empirical analysis to address the following research questions: **RQ1.** How do hallucinations and omissions vary with the insertion position of the clip and semantic similarity between the inserted clip and the original video? **RQ2.** Do narrative priors influence throughout the entire decoding process of Video LLMs? **RQ3.** How does reducing temporal context affect errors induced by narrative priors?

**Answer to RQ1: Narrative prior–induced errors depend on both insertion position and semantic similarity.** Figure 5 visualizes captioning metrics (EHR, EOR, IEOR) along the position and similarity axes. For Video-LLaVA, errors are strongly influenced by insertion position: end insertions frequently cause hallucinations and omissions of original events, while start and middle insertions more often lead to omission of inserted events. In contrast, LLaVA-OneVision shows stronger dependence on semantic similarity. When inserted clips closely resemble the target video, both hallucination and omission rates rise, and inserted clips are often ignored (high IEOR). At lower similarity levels, the model hallucinates and omits less—and in some cases even achieves lower omission rates than with the original video, as shown in Table 2.

**Answer to RQ2: Narrative priors influence the entire decoding process of Video LLMs.** To examine how narrative priors affect decoding, we measure the KL divergence between token-level probability distributions from (i) the original video $V_o$ and (ii) the composite video $V_c$:

$$D_{KL}^t = D_{KL}\big(p(y_t \mid V_o) \,\|\, p(y_t \mid V_c)\big), \tag{3}$$

where $y_t$ is the token generated at decoding step $t$. We average the KL divergence over 100 random samples to reduce noise and visualize results across normalized decoding steps $[0, 1]$ for Video-LLaVA and LLaVA-OneVision in Figure 6.

Video-LLaVA is mostly not influenced by inserted events to maintain narrative coherence, except when events are end inserted. Interestingly, end insertion also perturbs KL Divergence of earlier decoding steps more, compared to later steps. The results suggest that narrative priors shape the generation process from the beginning. Combined with the error patterns in Figure 5a, these results indicate that end insertions amplify narrative-prior effects, driving hallucinations and omissions. Note that high IEORs for start/mid insertions are largely explained by higher EHRs and EORs, since original events greatly outnumber inserted ones.

In contrast, LLaVA-OneVision exhibits higher KL divergence across all insertion positions, consistent with its reduced reliance on narrative priors. This aligns with its superior performance in Tables 1 and 2. However, for mid and end insertions, KL divergence also extends into entire decod-

| # Frames | LLaVA-NeXT-Video | | | mPLUG-Owl3 | | | LLaVA-OneVision-7B | | |
|---|---|---|---|---|---|---|---|---|---|
| | EHR | EOR | IEOR | EHR | EOR | IEOR | EHR | EOR | IEOR |
| 8 | 0.431 | 0.529 | 0.755 | 0.367 | 0.596 | 0.572 | 0.278 | 0.483 | 0.438 |
| 16 | 0.382 | 0.503 | 0.730 | 0.337 | 0.539 | 0.519 | 0.243 | 0.428 | 0.380 |
| 32 | 0.344 | 0.480 | 0.735 | 0.325 | 0.506 | 0.470 | 0.224 | 0.411 | 0.417 |
| 64 | 0.367 | 0.468 | 0.757 | 0.215 | 0.377 | 0.336 | - | - | - |

Table 3: Impact of temporal context on narrative prior–induced errors. We report event-level hallucination (EHR), omission (EOR), and inserted-event omission (IEOR) rates of LLaVA-NeXT-Video, mPLUG-Owl3, and LLaVA-OneVision-7B across different frame counts.

ing steps, implying that narrative priors in LLaVA-OneVision reshape the narrative throughout the entire generation process. For more details, please refer to Appendix C.

**Answer to RQ3: Limited temporal context amplifies reliance on narrative priors.** To test whether reduced temporal context strengthens reliance on narrative priors, we vary the number of input frames. With fewer frames, grounding signals weaken and models are expected to depend more on narrative continuity, increasing hallucinations and omissions.

Table 3 shows a consistent trend in LLaVA-NeXT-Video and mPLUG-Owl3: increasing frames reduces errors, confirming that stronger visual grounding alleviates them. Yet their behaviors diverge for IEOR. In LLaVA-NeXT-Video, omission stays high regardless of frame count, reflecting persistent suppression by narrative priors. In contrast, mPLUG-Owl3 shows a steady decline in omission with more frames, suggesting that stronger visual evidence offset narrative pressure.

Overall, these results indicate that reduced temporal context amplifies narrative-prior reliance, though the extent varies by architecture. Since current Video LLMs take sampled frames, such structural limitations make narrative prior–driven hallucinations and omissions unavoidable—underscoring the importance of diagnosing and addressing them.

## 4.4 DISCUSSION

Our findings show the critical role of narrative priors in shaping errors during video understanding. In our experiments, Video LLMs often produce outputs that conflict with visual evidence when trying to maintain narrative coherence, raising concerns for reliability-sensitive domains. For instance, an autonomous driving system might miss a sudden obstacle, like a wild animal crossing the road, if it breaks the expected flow of traffic. Similarly, a disaster-response robot could misinterpret or ignore unusual signals in chaotic environments, leading to poor situation assessment. These cases highlight the risks of narrative-driven errors and the need for systematic study of their impact.

Isolating errors caused purely by narrative priors is difficult. To address this, NOAH uses composite videos with inserted clips for controlled evaluation. This approach helps separate narrative effects but inevitably introduces artifacts less natural than real-world footage.

Yet many real videos—such as TV series with frequent scene changes or advertisements with rapid transitions—are themselves edited compositions of multiple events. Narrative-driven errors are therefore not confined to synthetic benchmarks but are prevalent in realistic scenarios. Future work should extend evaluation to more naturalistic corpora to capture this challenge more faithfully.

## 5 CONCLUSION

In this paper, we investigate hallucinations and omissions in Video LLMs driven by narrative priors. NOAH is a large-scale benchmark comprising composite videos with varying semantic similarity and insertion positions to isolate narrative-prior effects. Our experiments on both open- and closed-source models reveal that narrative priors consistently shape error patterns across architectures, similarity levels, and insertion positions, and that reducing temporal context further amplifies these effects. These results demonstrate that narrative priors are not occasional artifacts but a widespread and influential source of error in Video LLMs. By making these biases visible, NOAH establishes narrative priors as a central challenge for Video LLM reliability and offers the first standardized framework to diagnose and address them. We hope this benchmark will catalyze future research toward models that remain faithful to visual evidence while resisting narrative-driven distortions.

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

APPENDIX

## A    TASK DEFINITION

Building on the dataset design described above, our benchmark defines two complementary tasks that expose how narrative prior affects model predictions: (i) **Video Captioning under Narrative Priors**, and (ii) **Video Question Answering under Narrative Priors**. These tasks are designed to jointly capture generative and discriminative aspects of errors induced by narrative prior.

### A.1    VIDEO CAPTIONING UNDER NARRATIVE PRIORS.

A primary objective of open-ended video description is to generate captions that are factually faithful to the visual content of the video. The ability to produce an accurate and comprehensive description of all presented events is a critical measure of a model's reliability. To rigorously evaluate this core requirement, especially under conditions designed to induce narrative-driven errors, we propose a video captioning task. In this task, a model must generate a caption for a synthetic video, which is formed by inserting an additional event clip into a target video. This setup is designed to reveal a model's propensity to either hallucinate unsupported events or omit factual ones when constructing a contextually plausible narrative. We formally measure these effects using event-level hallucination and omission metrics presented in Section 4.1.

### A.2    VIDEO QUESTION ANSWERING UNDER NARRATIVE PRIORS.

In addition to open-ended captioning, we design binary question answering (QA) tasks to explicitly evaluate how narrative priors influence model predictions. These QA tasks are constructed on top of our curated dataset and are designed to test whether models can faithfully identify event existence, temporal order, and narrative plausibility under controlled settings. Accordingly, we introduce three complementary QA tasks as follows.

**Existence QA.** This task assesses a model's fundamental ability to determine the presence or absence of the inserted event clip. To construct a robust and balanced evaluation that is robust against trivial response biases, we formulate question pairs in both affirmative and negative formulations.

First, for the affirmative formulations, which query the presence of an event, we create a pair of questions following the template *"Is the event where {inserted event} present in the video?"*. One question uses the description of the inserted event, for which the ground truth is "Yes". The other uses the description of an absent distractor event, for which the ground truth is "No".

We then create the corresponding negative formulations for this pair by asking about absence, as in *"Is the event where {inserted event} absent in the video?"*. For these questions, the ground truths are logically inverted: the answer becomes "No" for the inserted event and "Yes" for the distractor. This comprehensive design ensures that a model is evaluated on its ability to ground answers in visual evidence, rather than relying on superficial cues in the question phrasing. The distractor events are randomly sampled from the ActivityNet-Captions dataset and verified to be absent through an LLM-based filtering process, ensuring the integrity of our negative samples.

**Temporal QA.** This task evaluates a model's ability to correctly identify the chronological order of the inserted event clip relative to its adjacent original event. To probe the model's understanding of sequential order, we formulate binary (Yes/No) questions following a template such as *"Before/After {reference event}, is the previous/next event {inserted event}?"*.

The specific 'reference event' used in the template depends on the insertion position, which determines the available adjacent context. When the inserted event has only one adjacent original event (i.e., at the start or end), we formulate a pair of questions against this single neighbor. For example, for insertions at the start, a question asking *"Before {succeeding event}, is the previous event {inserted event}?"* has a ground truth of "Yes". The corresponding question asking *"After {succeeding event}, is the next event {inserted event}?"* has a ground truth of "No".

When the inserted event is positioned in the middle and thus has two adjacent events, the reference event for the questions is chosen equally from both the preceding and the succeeding neighbors. This comprehensive approach allows for an analysis of how both forward and backward narrative contexts influence a model's temporal reasoning.

**Narrative QA.** This task evaluates a model's robustness against plausible but false narrative inferences that may be triggered by the inserted event. Unlike Existence QA, which tests for the presence of visually grounded facts, Narrative QA probes whether a model can distinguish between actual events and fabricated events that are contextually coherent with the surrounding narrative.

To achieve this, we formulate question pairs in both affirmative and negative formulations. For the affirmative formulations, which query the presence of an event, we create a pair of questions. One question asks about a factual event that is genuinely present in the video, for which the ground truth is "Yes". The other asks about a fabricated but plausible event, which is manually authored by humans to be narratively consistent but is absent from the video; for this query, the ground truth is "No".

We then create the corresponding negative formulations for this pair by asking about absence. In this case, the ground truths are logically inverted: the answer becomes "No" for the factual event and "Yes" for the fabricated event. This design allows us to assess if a model's reasoning is truly grounded in the video's visual content or if it is easily misled by plausible narrative fabrications.

# B  EXPERIMENTAL DETAILS

## B.1  EVALUATION PROMPTS AND API COSTS

For captioning evaluation, we assessed a total of 9,000 captions under two criteria: hallucination and omission. Each caption was evaluated once for each criterion, resulting in a total of 18,000 API requests.

Two judge models were employed for this evaluation:

- **GPT-4o**
- **Gemini 2.0 flash**

Based on the API pricing at the time of experiments, the total cost for processing all 18,000 requests was approximately $240 for GPT-4o and $20 for Gemini 2.0 flash.

The exact evaluation prompts used for hallucination and omission detection are provided in Figure 8 and Figure 9, respectively.

Each prompt follows a structured, step-by-step format with four stages: (1) event extraction from the caption or ground-truth, (2) application of hallucination/omission criteria, (3) event-by-event reasoning with justifications, and (4) final scalar metrics reported in a strict JSON schema. This design ensures consistency, reduces ambiguity in free-form model outputs, and enables automated parsing for metric computation.

## B.2  FRAME SAMPLING STRATEGY FOR CLOSED-SOURCE MODELS

For closed-source video LLMs such as GPT-4o and Gemini, we standardized the video input by extracting a fixed set of 128 frames per video. This ensures fair comparisons across models while keeping cost and runtime predictable.

**Uniform Frame Sampling.** We divide each video evenly into 128 segments and sample one frame per segment. This guarantees full temporal coverage from start to end, avoiding bias toward specific parts of the video. If the video has fewer than 128 frames, we use all available frames.

**Resolution Normalization.** Each sampled frame is converted to RGB and resized so that the longer side does not exceed 512 pixels, preserving the original aspect ratio. This keeps the visual content standardized and prevents excessive input size while maintaining sufficient detail.

**Encoding for API Requests.** Frames are compressed using JPEG (quality = 85) and converted to Base64 strings for transmission to closed-source APIs. This format balances efficiency, cost, and visual fidelity while allowing easy integration with model inputs.

**Fairness and Reproducibility.** All models receive the same number of frames, resolution, and compression settings, ensuring consistent inputs across evaluation runs. Deterministic sampling guarantees that identical videos always produce the same frame set.

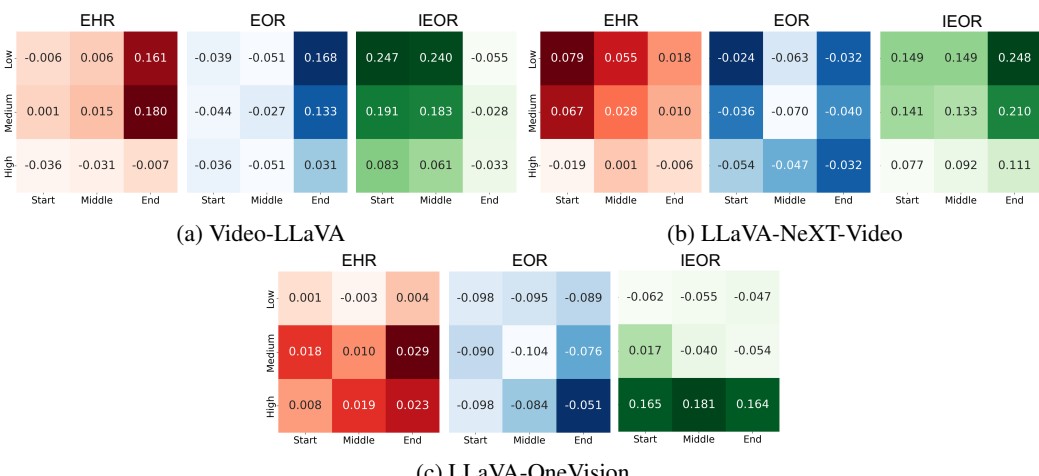

Figure 7: Delta Heatmaps of captioning metrics across similarity and insertion position.

# C    ADDITIONAL ANALYSIS

## C.1    ERROR FREQUENCY ANALYSIS

| Model | Hallucination (%) | | | Omission (%) | | | Inserted Omission (%) | | | Total H | Total O | Total IO |
|---|---|---|---|---|---|---|---|---|---|---|---|---|
| | – | 0 | + | – | 0 | + | – | 0 | + | | | |
| Video-LLaVA | 28.11 | 38.96 | 32.93 | 18.47 | 56.18 | 25.37 | 5.89 | 78.42 | 15.69 | 4.82 | 6.80 | 9.80 |
| LLaVA-OneVision | 32.82 | 28.99 | 38.19 | 28.40 | 50.40 | 21.20 | 3.33 | 56.93 | 39.73 | 5.36 | -7.20 | 36.40 |
| LLaVA-NeXT-Video | 32.78 | 29.03 | 38.19 | 28.40 | 50.40 | 21.20 | 8.60 | 67.52 | 23.88 | 5.41 | -7.20 | 15.27 |

Table 4: Error frequency analysis across models for hallucination, omission, and inserted omission. For each metric, the table reports the percentage of samples showing decrease (–), no change (0), or increase (+), along with the total frequency computed as $(V_c - V_o)$ or $(V_c - V_i)$ over 9000 captions.

Table 4 reports the error frequency analysis across all models for hallucination, omission, and inserted omission. Here, frequencies are computed as differences between outputs with narrative context and baselines $(V_c - V_o)$ or, for inserted omission, between outputs with context and those containing artificially inserted events $(V_c - V_i)$, thus reflecting **error frequencies induced by narrative priors**. We observe that **inserted omission errors exhibit a pronounced increase across all models**, with the majority of samples falling into the "+" category, indicating that artificially inserted events are often missed under narrative priors. **Hallucination errors show a similar trend**, with increases outweighing decreases for most models, suggesting that narrative context encourages additional, sometimes spurious, content. In contrast, **omission errors tend to decrease** (negative values), implying that the presence of narrative context can help reduce the omission of ground-truth events. Overall, these results highlight distinct behavioral patterns for each error type under narrative priors.

## C.2    DELTA HEATMAPS

Figure 7 presents delta heatmaps that visualize changes in hallucination (EHR), omission (EOR), and inserted-event omission (IEOR) across different similarity and insertion settings. Compared with the baseline ($V_o$ or $V_i$), positive values indicate an increase in errors under composite videos ($V_c$), while negative values denote a decrease. These heatmaps complement the main results by illustrating how error frequencies vary at a finer granularity across narrative prior conditions.

## C.3    GEMINI EVALUATION RESULTS AND CORRELATION WITH GPT

Table 5 summarizes the evaluation results obtained using Gemini 2.0 flash as the evaluator across all benchmark metrics, following the same format as the GPT-based results reported in the main paper.

| Models | # LLM Params | #F | Captioning | | | QA | | |
|---|---|---|---|---|---|---|---|---|
| | | | CHR / EHR (↓) | COR / EOR (↓) | IEOR (↓) | Exist. (↑) | Temp. (↑) | Narr. (↑) |
| **Open-source video LLMs** | | | | | | | | |
| BLIP-3-Video | 4B | 8 | 0.779 / 0.379 | 0.960 / 0.447 | 0.597 | 0.364 | 0.101 | 0.142 |
| Video-LLaVA | 7B | 8 | 0.706 / 0.467 | 0.982 / 0.604 | 0.798 | 0.290 | 0.039 | 0.276 |
| LLaVA-NeXT-Video | 7B | 8 | 0.753 / 0.431 | 0.977 / 0.529 | 0.755 | 0.387 | 0.100 | 0.281 |
| LLaVA-OneVision | 7B | 8 | 0.593 / 0.199 | 0.924 / 0.407 | 0.335 | 0.685 | 0.486 | 0.387 |
| LongVA | 7B | 16 | 0.610 / 0.269 | 0.914 / 0.348 | 0.533 | 0.041 | 0.260 | 0.029 |
| mPLUG-Owl3 | 7B | 16 | 0.643 / 0.228 | 0.932 / 0.397 | 0.389 | 0.781 | 0.499 | 0.212 |
| ST-LLM | 7B | 64 | 0.714 / 0.335 | 0.967 / 0.443 | 0.607 | 0.250 | 0.060 | 0.283 |
| Video-LLaMA3 | 2B | 128 | 0.683 / 0.232 | 0.907 / 0.333 | 0.425 | 0.348 | 0.341 | 0.304 |
| Video-LLaMA3 | 7B | 128 | 0.555 / 0.171 | 0.891 / 0.313 | 0.374 | 0.460 | 0.460 | 0.154 |
| Qwen2.5-VL | 3B | 128 | 0.682 / 0.234 | 0.898 / 0.352 | 0.343 | 0.680 | 0.345 | 0.356 |
| Qwen2.5-VL | 7B | 128 | 0.608 / 0.163 | 0.849 / 0.302 | 0.286 | 0.970 | 0.600 | 0.536 |
| Qwen2.5-VL | 32B | 128 | 0.769 / 0.202 | 0.653 / 0.187 | 0.188 | 0.973 | 0.655 | 0.592 |
| Qwen2.5-VL | 72B | 128 | 0.642 / 0.183 | 0.863 / 0.324 | 0.277 | 0.937 | 0.682 | 0.796 |
| **Closed-source video LLMs** | | | | | | | | |
| Gemini 2.5 Flash | - | 128 | 0.478 / 0.088 | 0.383 / 0.093 | 0.111 | 0.686 | 0.597 | 0.668 |
| GPT–4o | - | 128 | 0.414 / 0.107 | 0.751 / 0.192 | 0.325 | 0.703 | 0.148 | 0.471 |

Table 5: Evaluation results on NOAH with Gemini 2.0 Flash as the evaluator. We report the performance of open- and closed-source Video LLMs on captioning and QA tasks. QA performance is measured by accuracy on Existence (Exist.), Temporal (Temp.), and Narrative (Narr.) questions.

| Metric | CHR | EHR | COR | EOR | IEOR | ALL |
|---|---|---|---|---|---|---|
| **Pearson** | 0.8496 | 0.9696 | 0.9878 | 0.9893 | 0.9901 | 0.9835 |
| **Spearman** | 0.7857 | 0.9580 | 0.9955 | 0.9786 | 0.9929 | 0.9854 |

Table 6: Correlation analysis between Gemini and GPT evaluation results across metrics.

This parallel presentation allows for a direct comparison between the two evaluators under identical experimental settings.

To further quantify the agreement between Gemini 2.0 flash and GPT-4o, we compute both **Pearson** and **Spearman** correlation coefficients across all metrics, as shown in Table 6. Pearson correlation measures the linear relationship between the two evaluators, while Spearman correlation captures the rank-based consistency, offering a complementary perspective on evaluation reliability.

Table 6 shows that most correlations remain above $0.95$, with the lowest (CHR Spearman) still above $0.78$ and several metrics—including COR, EOR, and IEOR—achieving correlations close to or above $0.99$. Even when aggregating all individual scores across metrics, the overall correlations remain exceptionally high (Pearson $0.98+$, Spearman $0.98+$). These findings confirm that **Gemini and GPT exhibit highly consistent evaluation behavior** across all hallucination and omission metrics, indicating that either evaluator can provide reliable and interchangeable assessments for the benchmark tasks.

You are evaluating a video caption for event hallucination errors.

Your task is to determine whether any events in the inference caption describe information that is not supported by the ground truth events.

Please follow the steps carefully, and format your output exactly as specified.

──────────── EVALUATION STEPS ────────────
**STEP 1 – EXTRACTED_EVENTS**
- Review the inference caption and extract only events that describe concrete agent-action-object interactions
- Each event must capture a meaningful action performed by an agent involving a key object or participant.
- Number the resulting events (1., 2., ...)

Do NOT extract any of the following as events:
- Static background descriptions (e.g., crowd, tools, weather, lighting, location, structural surroundings)
- Spatial relationships and static configurations (e.g., "the court is surrounded by a fence", "the room has windows", "there are chairs around the table")
- Existential statements about objects or settings (e.g., "there is a lamp", "the area contains equipment", "a building stands nearby")
- Minor attributes (e.g., color, clothing, age, time of day)
- Text content, on-screen overlays, banners, labels, captions, or subtitles
- Any references to reading, seeing, or describing visible text
- Stylistic or emotional expressions (e.g., "focused expression", "dramatic moment")
- Paraphrased general impressions or vague summaries (e.g., "showcases skills", "demonstrates technique")
- Visual conditions without agent interaction (e.g., "the lighting is dim", "a ladder is in the background")
- Any emotional or affective states inferred from appearance (e.g., "looked happy", "seemed determined", "appears moved")

**STEP 2 – HALLUCINATION CRITERIA**
- An event is defined as an **argument structure** including participants (agents/objects) and actions. An event is considered **hallucinated** if it introduces new factual content that cannot be reasonably supported by ground truth events.

- We evaluate hallucination at the **event level** based on semantic consistency. An event includes a combination of actions, agents, and key objects that form its core meaning. We do not assess individual words or isolated details.

- Even if certain descriptive details are described differently (e.g., colors, sizes, emotional states), the event is not considered hallucinated as long as the core participants (agents/objects) and actions clearly match those in the ground truth events, with no substitutions (e.g., different materials, objects, or action types).

- We focus on detecting:
    1. **New actions**: Actions or behaviors not present or inferable from ground truth
    2. **New participants**: Agents or objects introduced without ground truth support
    3. **Contradictory content**: Information that directly conflicts with ground truth events

- Background elements and minor additions of **attributes** (e.g., color, clothing, environmental details, atmospheric conditions) do NOT count as hallucinations — unless they contradict ground truth or fundamentally introduce new event content.

- Stylistic interpretations, emotional descriptions, reasonable visual inferences, paraphrasing, and variations in tone are not hallucinations.

- Ignore any mentions of text content, overlays, or subtitles when extracting events, and do not count them as hallucinations.

For each extracted event, determine whether it is supported by ground truth content:
- **Explicit** – Clearly and directly stated in ground truth, with the same core components (agents/objects/actions)
- **Implicit** – Strongly implied or logically inferable from ground truth, with the same core components (agents/objects/actions)
- **Paraphrased** – Reworded but semantically equivalent to ground truth content, with the same core components (agents/objects/actions)
- **Visually synonymous** – Different wording is allowed only if the visual event is clearly the same and all core components (agents, objects, actions) remain identical with no substitutions.

If an event is not supported by any of the above, mark it as hallucinated.

**STEP 3 – EVENT-BY-EVENT REASONING**
- For each extracted event, write a brief sentence following this format:
    • Event #{N} – This event — [short description] — was [SUPPORTED / HALLUCINATED] because [detailed justification].

- Justify your judgment based on the hallucination criteria from Step 2.

**STEP 4 – FINAL DETECTION**
Provide the conclusive metric only:
- HALLUCINATION_COUNT: [number of hallucinated events]
────────────────────────────────────

──────────── OUTPUT FORMAT (strictly follow) ────────────

EXTRACTED_EVENTS:
1. [Extracted event 1]
2. [Extracted event 2]
...

CRITERIA_REVIEW:
- Apply the hallucination criteria defined in STEP 2 above.

EVENT-BY-EVENT REASONING:
• Event #1 – This event — [short description] — was [SUPPORTED / HALLUCINATED] because [detailed justification].
• Event #2 – ...
• Event #3 – ...
...

FINAL METRICS:
- HALLUCINATION_COUNT: [number of hallucinated events]
────────────────────────────────────

────────────────────────────────────

GROUND-TRUTH EVENTS (reference for evaluation):
{GROUND_TRUTH_EVENTS}

INFERENCE CAPTION:
"{INFERENCE_CAPTION}"
────────────────────────────────────

Figure 8: Hallucination evaluation prompt

You are evaluating a video caption for event omission errors.

Your task is to determine whether any ground truth events are omitted from the caption.

Some ground-truth events may have been artificially inserted for evaluation purposes. These are referred to as "inserted events."

Please follow the steps carefully, and format your output exactly as specified.

———————— EVALUATION STEPS ————————
**STEP 1 – GROUND_TRUTH_EVENTS**
- Review both the Ground Truth Caption and the Supporting Visual Description for each event.
- Based on these, extract all events as discrete semantic units that capture meaningful actions, agents, and interactions.
- For each event, write a representative caption focusing on its main action, agent, and key object. Use the supporting visual description only to clarify visual content, not abstract context.
- Number the resulting events (1., 2., ...), and clearly mark which ones were artificially inserted for evaluation purposes.

**STEP 2 – OMISSION CRITERIA**
- An event is defined as an **argument structure** including participants (agents/objects) and actions. An event is considered **omitted** if core components are missing such that the event's meaning cannot be sufficiently conveyed — either explicitly, implicitly, through paraphrasing, or visual synonymy.

- Even if certain components of the original event are not explicitly mentioned, the event is not considered omitted as long as its core meaning is preserved and reasonably inferable.

- We evaluate omission at the **event level** based on semantic completeness. An event includes a combination of actions, agents, and key objects that form its core meaning. We do not assess individual words or isolated details.

- **For events whose core meaning contains multiple actions**: If an event's core meaning includes several actions (e.g., "A person wipes, measures, and rolls paper"), the event is considered **INFERRED** if **any one** of those actions is properly described in the caption. Complete omission occurs only when **none** of the actions are mentioned or inferable.

- **Peripheral participants** (e.g., onlookers, bystanders, crowd members, people watching) are **NOT** required for an event to be considered complete. Their absence does **NOT** constitute an omission.

- Minor omissions of **attributes** (e.g., color, age, time, clothing) do NOT count as omissions — unless the missing attribute fundamentally alters the identity or core purpose of the event.

- Incorrect mentions are considered hallucinations, not omissions. If an event is referenced but contains factual errors — such as wrong actions, agents, or objects — it is not an omission but a hallucination.

- Ignore stylistic differences, paraphrasing, or variation in tone — only assess whether the event is **faithfully represented** in meaning.

- We track two types of omissions:
  1. **Total omission**: Ground-truth events that are not mentioned or inferable in the caption.
  2. **Inserted event omission**: Among them, the subset of omitted events that were inserted.

For each ground-truth event, determine whether it is faithfully represented in the caption:
- A ground-truth event is considered **present (INFERRED)** if its core action and agent are reasonably conveyed — either:
  • **Explicitly** – Clearly and directly stated
  • **Implicitly** – Strongly implied or logically inferable
  • **Paraphrased** – Reworded but semantically equivalent
  • **Visually synonymous** – Described differently but visually equivalent (e.g., "a man takes a bite" vs. "a man eats")
- For events whose **core meaning contains multiple actions**, the event is **INFERRED** if **any one** of those actions is properly represented.
- An event is considered **OMITTED** only if **none** of its actions (within the core meaning) are inferable by any of the above methods.
- Supporting Visual Descriptions may assist your understanding, but should not be treated as required evidence.

**STEP 3 – EVENT-BY-EVENT REASONING**
- For each ground-truth event, write a brief sentence following this format:
  • Event #{N} – This event — [short description] — was [INFERRED / OMITTED] because [detailed justification]. This event [was / was not] an inserted one.

- Justify your judgment based on the omission criteria from Step 2.

**STEP 4 – FINAL DETECTION**
Provide the conclusive metric only:
- TOTAL_OMISSION_COUNT: [number of all omitted events]
- INSERTED_OMISSION_COUNT: [number of omitted events that were inserted]
————————————————————————————

———————— OUTPUT FORMAT (strictly follow) ————————

GROUND_TRUTH_EVENTS:
1. [Ground-truth event 1]
2. [Ground-truth event 2]
...

CRITERIA_REVIEW:
- An event is counted as OMITTED only if its **core meaning** (main action + agent + key object) is not recoverable via explicit, paraphrased, implied, or visually synonymous cues in the caption.
- For events whose **core meaning contains multiple actions**, the event is **INFERRED** if **any one** of those actions is properly described.

EVENT-BY-EVENT REASONING:
• Event #1 – This event — [short description] — was [INFERRED / OMITTED] because [justification]. This event [was / was not] an inserted one.
• Event #2 – ...
• Event #3 – ...
...

FINAL METRICS:
- TOTAL_OMISSION_COUNT: [number of all omitted events]
- INSERTED_OMISSION_COUNT: [number of omitted events that were inserted]
————————————————————————————

————————————————————————————

GROUND-TRUTH EVENTS (extracted from ground truth caption and visual description):
{GROUND_TRUTH_EVENTS}

INSERTED EVENT to check (at position {INSERT_POSITION}):
"{INSERTED_EVENT}"

INFERENCE CAPTION:
"{INFERENCE_CAPTION}"
————————————————————————————

Figure 9: Omission evaluation prompt

