# OpenReview forum: "NOAH: Benchmarking Narrative Prior driven Hallucination and Omission in Video Large Language Models"
_ICLR.cc/2026/Conference — ICLR 2026 Conference Withdrawn Submission_

### Official Review · Reviewer_1UjX · 2025-10-26

**Soundness:** 4
**Presentation:** 3
**Contribution:** 2
**Rating:** 4
**Confidence:** 5

**Summary:**

The paper introduces a new benchmark to evaluate how narrative priors in Video Large Language Models (Video LLMs) cause hallucinations and omissions. The benchmark is curated by inserting random events at the beginning, middle, and end of videos, and is accompanied by metrics, such as Inserted Event Omission Rate and Event Hallucination Rate.

**Strengths:**

•	This is the first benchmark that explicitly investigates the impact of narrative priors in Video LLMs, which is a valuable contribution to the research community.

•	The benchmark is technically sound and well-constructed.

**Weaknesses:**

•	My main concern is that the paper’s contributions are limited. As reported in lines 315–317, all methods already show high hallucination and omission rates on the original data, so it’s expected they would perform similarly on videos with inserted events. From my viewpoint, the main findings are restricted to (i) a method who performs better in hallucination and omission than other may behave differently on the new benchmark, and (ii) how the temporal position of inserted events affects final performance.

•	The paper presentation needs improvement. For example:

o	It lacks visual examples from the dataset.

o	In lines 315–321, results are discussed without referencing the specific table or figure.

o	They do not clearly mention Table 1 in the text.

o	There is a need for clearer explanation of the difference between Vi and Vc. Vi has 8 frames, while Vc has 9—how do Vo and Vi have the same frame count if Vi includes inserted frames?

**Questions:**

Please see the weaknesses part.

**Details Of Ethics Concerns:**

There is no Etichs Statment section in the paper, and do they release the data?

---

### Official Review · Reviewer_5pBM · 2025-10-29

**Soundness:** 2
**Presentation:** 2
**Contribution:** 2
**Rating:** 2
**Confidence:** 5

**Summary:**

This paper introduces a benchmark assessing hallucinations and omissions derived by narrative prior of Video-LLMs. The benchmark dataset is curated by inserting irrelevant video clips between certain events, and validates if the model can recognize both false positives and false negatives. Throughout experiments, this paper claims that narrative prior induces hallucinations and omissions of Video LLMs, and the effect varies by semantic similarity and position of inserted frames.

**Strengths:**

- This paper proposes a large scale video-QA benchmark with 9k synthetically composed videos.
- The benchmark offers diverse (8 unique) metrics from all possible failure scenarios.
- RQ2 and RQ3 in Section 4.3 provides new findings to the community.

**Weaknesses:**

- I didn’t fully understand the concept of narrative prior-induced hallucinations. Definition of narrative prior from the authors can be rephrased to: “inductive bias of Video LLMs to generate contextually accurate captions grounded by visual evidence”. Then does it mean that Video-LLMs have a bias to ignore contextually irrelevant frames/clips of the video? This paper lacks an explanation why models have such bias, and assumes this prior to be true without proof of concept. Meanwhile, it would be better to simply denote it as visual cognition/perception.
- Detailed elaboration of the human validation process for AI-generated questions is missing.
- Exactly how are inaccuracies induced from narrative prior distinct from existing literature of general visual cognition benchmarks? VidHallucer identified intrinsic and extrinsic hallucination categories with similar intuition, Video-MME assessed perception, cognition, and synopsis capabilities, VidHalluc diagnosed hallucinations with paired videos derived from visual encoders with different inductive bias, and ARGUS analyzed hallucination and omission effects on Video-LLMs. I cannot see any novel contributions distinct from these benchmarks.


I will increase the score once current concerns are addressed.

**Questions:**

- Why 3 groups of similarity scores in Figure 3 have all bell shaped curves for each category? Intuitively, it shall be overall uniformly or normally distributed. Is it a coincidence or deliberately separated by this distribution?
- Authors concluded that “narrative priors drive hallucinations and omissions in Video LLMs” at line 344. I believe this is a contradiction: how can a prior to “generate contextually accurate captions grounded by visual evidence” induce failures? If so, should we train models with a dataset that invalidates the narrative prior? Also, analysis on the cause of the failures induced by narrative priors is missing.
- Is the answer to the research question in line 346 the subsequent paragraph? Or is it missing?
- Paragraph in line 363 describes that LLaVA families show better results when inserted in other video clips. Exactly which architecture triggered this phenomenon? What kind of video clips encourage this performance gain? Can we generalize the EOR improvement on LLaVA-NeXT-Video and LLaVA-OneVision to improve overall Video-LLM performance? The claim is not convincing if these questions are not addressed.
- Paragraph at line 376 that models successfully caption $V_i$ but fails when combined as $V_c$. Isn’t it because the other frames acted as a distractor? I believe analysis of attention maps toward $V_i$ might give a clue.
- Does “decoding steps [0,1]” at line 421 indicate each hidden layer?

---

### Official Review · Reviewer_4ivZ · 2025-10-31

**Soundness:** 2
**Presentation:** 2
**Contribution:** 2
**Rating:** 2
**Confidence:** 3

**Summary:**

This article addresses the "narrative prior" problem in Video LLMs, where the pursuit of narrative coherence leads to neglecting visual evidence, by proposing a benchmark named NOAH. The study finds: Current Video LLMs widely exhibit hallucination and omission problems caused by narrative prior; the model's generation of hallucinations and omissions is related to model architecture, the semantic similarity of inserted clips, and the insertion position; reducing the number of input frames strengthens the model's reliance on narrative prior, and it is confirmed through the decoding process that this bias systematically affects the output distribution from the very beginning of generation.

**Strengths:**

1.	The problem perspective is novel, for the first time systematically attributing the hallucination and omission problems of Video LLMs to a fundamental bias, namely "narrative prior".
2.	The paper's evaluation system design is comprehensive, setting up one captioning task and three QA tasks to evaluate the model from multiple dimensions: Existence, Temporal, and Narrative.

**Weaknesses:**

1.	This study bases its evaluation on another LLM system that also has inherent flaws of hallucination and omission, so its evaluation criteria themselves cannot be precisely verified.
2.	The generation method for composite videos is unnatural. The model's errors might reflect its lack of robustness to clipping behavior, rather than a true "narrative prior".
3.	When the model omits an inserted clip on NOAH, the paper interprets this as the model actively ignoring it to maintain narrative consistency. However, an equally convincing explanation is that the model's spatio-temporal attention mechanism failed to capture this isolated, context-irrelevant visual signal. The NOAH experiments fail to rule out the latter possibility, therefore its conclusion that "narrative prior" is the dominant factor causing the errors is not logically rigorous.
4.	The paper discovers the "narrative prior" problem of video large models from a novel perspective, but it does not discuss its root cause. Conducting some reasonable theoretical analysis here would help make the "narrative prior" concept more convincing.

**Questions:**

1.	There is uncertainty as to whether the average cosine similarity of static images can fully capture the narrative correlation of video time. Can you further verify the effectiveness of using the average cosine similarity of frames' CLIP embeddings as the similarity control method?
2.	Large language models themselves are known to have errors such as hallucinations and omissions. Can you provide what measures you used to ensure the reliability of using Gemini in the refinement process?

---

### Official Review · Reviewer_PMNJ · 2025-11-01

**Soundness:** 2
**Presentation:** 3
**Contribution:** 2
**Rating:** 4
**Confidence:** 4

**Summary:**

The paper presents NOAH, a new benchmark for evaluating narrative prior-driven hallucination and omission in Video Large Language Models. The benchmark provides a dataset of 9,000 composite videos, created by systematically inserting event clips into target videos with varying semantic similarity and insertion positions, yielding over 60,000 evaluation samples. It aims to systematically analyze how a model's inductive bias for narrative coherence leads to factual errors, such as fabricating events (hallucination) or suppressing them (omission). Several distinct tasks are defined using this dataset: a video captioning task with tailored metrics, and three specific question-answering tasks (Existence, Temporal, and Narrative QA). The paper also presents an evaluation of numerous baseline models, demonstrating that narrative-driven errors are a significant and widespread challenge for current Video LLMs.

**Strengths:**

1.	NOAH introduces a novel benchmark to evaluate MLLM hallucinations and omissions, which is critical for video understanding.
2.	The paper benchmarks 15 MLLMs, featuring extensive analysis of narrative prior placement and difficulty levels.

**Weaknesses:**

1.	 Insufficient Combination Granularity: Relying on only nine composite types provides a coarse benchmark, making it difficult to sufficiently analyze the granular conditions under which hallucinations and omissions occur.
2.	Limited Temporal Scope: The benchmark's limited time range (video duration) prevents exploration of how sequence length interacts with and potentially affects the model's utilization of narrative priors.
3.	Inadequacy of CLIP Score: The reliance on CLIP score is a limitation, as this metric is often insufficient for accurately measuring the high-level semantic similarity and temporal consistency specific to video content.
4.	Lack of Diversity and Scalability: The benchmark's construction from a single source (ActivityNet-Captions) and its limitation to only three evaluation tasks severely restricts its diversity and scalability. A more robust benchmark would require more varied data sources and a more comprehensive suite of evaluation tasks.
5.	Lack of some visualizations for the different composite video.

**Questions:**

1.	Disentangling Modality Contributions: Can the study determine if the narrative prior is derived from visual information or textual context? An experiment replacing video frames with text captions could help isolate the source.
2.	Impact of "Unnatural" Artifacts: Is it possible that the models are failing simply because they cannot process the "unnatural" artifacts of the composite video, rather than failing the intended narrative task?
3.	Architectural Basis for Performance Gaps: The results show clear performance differences (e.g., LLaVA-OneVision's robustness). What are the potential architectural reasons underpinning why some models are less prone to these errors than others?

---

### Note · Authors · 2025-11-13

**Comment:**

We respectfully request the withdrawal of our submission. We sincerely appreciate the reviewers and area chairs for the time, effort, and thoughtful feedback provided during the evaluation process.  Thank you again for your support and consideration.

**Withdrawal Confirmation:**

I have read and agree with the venue's withdrawal policy on behalf of myself and my co-authors.